# Data-Efficient Reinforcement Learning in Continuous State-Action Gaussian-POMDPs

**Rowan Thomas McAllister**
Department of Engineering
Cambridge University
Cambridge, CB2 1PZ
rtm26@cam.ac.uk

**Carl Edward Rasmussen**
Department of Engineering
University of Cambridge
Cambridge, CB2 1PZ
cer54@cam.ac.uk

## Abstract

We present a data-efficient reinforcement learning method for continuous state-action systems under significant observation noise. Data-efficient solutions under small noise exist, such as PILCO which learns the cartpole swing-up task in 30s. PILCO evaluates policies by planning state-trajectories using a dynamics model. However, PILCO applies policies to the observed state, therefore planning in *observation space*. We extend PILCO with filtering to instead plan in *belief space*, consistent with partially observable Markov decisions process (POMDP) planning. This enables data-efficient learning under significant observation noise, outperforming more naive methods such as *post-hoc* application of a filter to policies optimised by the original (unfiltered) PILCO algorithm. We test our method on the cartpole swing-up task, which involves nonlinear dynamics and requires nonlinear control.

## 1  Introduction

The Probabilistic Inference and Learning for COntrol (PILCO) [5] framework is a reinforcement learning algorithm, which uses Gaussian Processes (GPs) to learn the dynamics in continuous state spaces. The method has shown to be highly *efficient* in the sense that it can learn with only very few interactions with the real system. However, a serious limitation of PILCO is that it assumes that the observation noise level is small. There are two main reasons which make this assumption necessary. Firstly, the dynamics are learnt from the noisy observations, but learning the transition model in this way doesn't correctly account for the noise in the observations. If the noise is assumed small, then this will be a good approximation to the real transition function. Secondly, PILCO uses the noisy observation directly to calculate the action, which is problematic if the observation noise is substantial. Consider a policy controlling an unstable system, where high gain feed-back is necessary for good performance. Observation noise is *amplified* when the noisy input is fed directly to the high gain controller, which in turn injects noise back into the state, creating cycles of increasing variance and instability.

In this paper we extend PILCO to address these two shortcomings, enabling PILCO to be used in situations with substantial observation noise. The first issue is addressed using the so-called *Direct* method for training the transition model, see section 3.3. The second problem can be tackled by *filtering* the observations. One way to look at this is that PILCO does planning in observation space, rather than in belief space. In this paper we extend PILCO to allow filtering of the state, by combining the previous state distribution with the dynamics model and the observation using Bayes rule. Note, that this is easily done when the controller is being applied, but to gain the full benefit, we have to also take the filter into account when optimising the policy.

PILCO trains its policy through minimising the expected predicted loss when *simulating* the system and controller actions. Since the dynamics are not known exactly, the simulation in PILCO had to

simulate *distributions* of possible trajectories of the physical state of the system. This was achieved using an analytical approximation based on moment-matching and Gaussian state distributions. In this paper we thus need to augment the simulation over physical states to include the state of the filter, an *information state* or *belief state*. A complication is that the belief state is itself a probability distribution, necessitating simulating distributions over distributions. This allows our algorithm to not only apply filtering during execution, but also anticipate the effects of filtering during training, thereby learning a better policy.

We will first give a brief outline of related work in section 2 and the original PILCO algorithm in section 3, including the proposed use of the 'Direct method' for training dynamics from noisy observations in section 3.3. In section 4 will derive the algorithm for POMDP training or planning in belief space. Note an assumption is that we observe noisy versions of the state variables. We do not handle more general POMDPs where other unobserved states are also learnt nor learn any other mapping from the state space to observations other than additive Gaussian noise. In the final sections we show experimental results of our proposed algorithm handling observation noise better than competing algorithms.

## 2    Related work

Implementing a filter is straightforward when the system dynamics are *known* and *linear*, referred to as Kalman filtering. For known nonlinear systems, the extended Kalman filter (EKF) is often adequate (e.g. [13]), as long as the dynamics are approximately linear within the region covered by the belief distribution. Otherwise, the EKF's first order Taylor expansion approximation breaks down. Larger nonlinearities warrant the unscented Kalman filter (UKF) – a deterministic sampling technique to estimate moments – or particle methods [7, 12]. However, if moments can be computed analytically and exactly, moment-matching methods are preferred. Moment-matching using distributions from the exponential family (e.g. Gaussians) is equivalent to optimising the Kullback-Leibler divergence $\mathrm{KL}(p||q)$ between the true distribution $p$ and an approximate distribution $q$. In such cases, moment-matching is less susceptible to model bias than the EKF due to its conservative predictions [4].

Unfortunately, the literature does not provide a continuous state-action method that is both data efficient and resistant to noise when the dynamics are *unknown* and *locally nonlinear*. Model-free methods can solve many tasks but require thousands of trials to solve the cartpole swing-up task [8], opposed to model-based methods like PILCO which requires about six. Sometimes the dynamics are partially-known, with known functional form yet unknown parameters. Such 'grey-box' problems have the aesthetic solution of incorporating the unknown dynamics parameters into the state, reducing the learning task to a POMDP planning task [6, 12, 14]. Finite state-action space tasks can be similarly solved, perhaps using Dirichlet parameters to model the finitely-many state-action-state transitions [10]. However, such solutions are not suitable for continuous-state 'black-box' problems with no prior dynamics knowledge. The original PILCO framework does not assume task-specific prior dynamics knowledge (only that the prior is vague, encoding only time-independent dynamics and smoothness on some unknown scale) yet assumes full state observability, failing under moderate sensor noise. One proposed solution is to filter observations during policy execution [4]. However, without also predicting system trajectories w.r.t. the filtering process, a policy is merely optimised for unfiltered control, not filtered control. The mismatch between unfiltered-prediction and filtered-execution restricts PILCO's ability to take full advantage of filtering. Dallaire et al. [3] optimise a policy using a more realistic filtered-prediction. However, the method neglects model uncertainty by using the maximum a posteriori (MAP) model. Unlike the method of Deisenroth and Peters [4] which gives a full probabilistic treatment of the dynamics predictions, work by Dallaire et al. [3] is therefore highly susceptible to model error, hampering data-efficiency.

We instead predict system trajectories using closed loop filtered control precisely because we execute closed loop filtered control. The resulting policies are thus optimised for the specific case in which they are used. Doing so, our method retains the same data-efficiency properties of PILCO whilst applicable to tasks with high observation noise. To evaluate our method, we use the benchmark cartpole swing-up task with noisy sensors. We show that realistic and probabilistic prediction enable our method to outperform the aforementioned methods.

---

**Algorithm 1** PILCO

---

1: *Define* policy's functional form: $\pi : z_t \times \psi \rightarrow u_t$.
2: *Initialise* policy parameters $\psi$ randomly.
3: **repeat**
4:     *Execute* policy, record data.
5:     *Learn* dynamics model $p(f)$.
6:     *Predict* state trajectories from $p(X_0)$ to $p(X_T)$.
7:     *Evaluate* policy:     $J(\psi) = \sum_{t=0}^{T} \gamma^t \mathcal{E}_t$,     $\mathcal{E}_t = \mathbb{E}_X[\text{cost}(X_t)|\psi]$.
8:     *Improve* policy:     $\psi \leftarrow \text{argmin}_\psi J(\psi)$.
9: **until** policy parameters $\psi$ converge

---

## 3  The PILCO algorithm

PILCO is a model-based policy-search RL algorithm, summarised by Algorithm 1. It applies to continuous-state, continuous-action, continuous-observation and discrete-time control tasks. After the policy is executed, the additional data is recorded to train a probabilistic dynamics model. The probabilistic dynamics model is then used to predict one-step system dynamics (from one timestep to the next). This allows PILCO to probabilistically predict multi-step system trajectories over an arbitrary time horizon $T$, by repeatedly using the predictive dynamics model's output at one timestep, as the (uncertain) input in the following timestep. For tractability PILCO uses moment-matching to keep the latent state distribution Gaussian. The result is an analytic distribution of state-trajectories, approximated as a joint Gaussian distribution over $T$ states. The policy is evaluated as the expected total cost of the trajectories, where the cost function is assumed to be known. Next, the policy is improved using local gradient-based optimisation, searching over policy-parameter space. A distinct advantage of moment-matched prediction for policy search instead of particle methods is smoother policy gradients and fewer local optima [9]. This process then repeats a small number of iterations before converging to a locally optimal policy. We now discuss details of each step in Algorithm 1 below, with policy evaluation and improvement discussed Appendix B.

### 3.1  Execution phase

Once a policy is initialised, PILCO can *execute* the system (Algorithm 1, line 4). Let the latent state of the system at time $t$ be $x_t \in \mathbb{R}^D$, which is noisily observed as $z_t = x_t + \epsilon_t$, where $\epsilon_t \overset{iid}{\sim} \mathcal{N}(0, \Sigma^\epsilon)$. The policy $\pi$, parameterised by $\psi$, takes observation $z_t$ as input, and outputs a control action $u_t = \pi(z_t, \psi) \in \mathbb{R}^F$. Applying action $u_t$ to the dynamical system in state $x_t$, results in a new system state $x_{t+1}$. Repeating until horizon $T$ results in a new single state-trajectory of data.

### 3.2  Learning dynamics

To learn the unknown dynamics (Algorithm 1, line 5), any probabilistic model flexible enough to capture the complexity of the dynamics can be used. Bayesian nonparametric models are particularly suited given their resistance to overfitting and underfitting respectively. Overfitting otherwise leads to model bias - the result of optimising the policy on the erroneous model. Underfitting limits the complexity of the system this method can learn to control. In a nonparametric model no prior dynamics knowledge is required, not even knowledge of *how complex* the unknown dynamics might be since the model's complexity grows with the available data. We define the latent dynamics $f : \tilde{x}_t \rightarrow x_{t+1}$, where $\tilde{x}_t \doteq [x_t^\top, u_t^\top]^\top$. PILCO models the dynamics with $D$ independent Gaussian process (GP) priors, one for each dynamics output variable: $f^a : \tilde{x}_t \rightarrow x_{t+1}^a$, where $a \in [1, D]$ is the $a$'th dynamics output, and $f^a \sim \mathcal{GP}(\phi_a^\top \tilde{x}, k^a(\tilde{x}_i, \tilde{x}_j))$. Note we implement PILCO with a linear mean function[1], $\phi_a^\top \tilde{x}$, where $\phi_a$ are additional hyperparameters trained by optimising the marginal likelihood [11, Section 2.7]. The covariance function $k$ is squared exponential, with length scales $\Lambda_a = \text{diag}([l_{a,1}^2, ..., l_{a,D+F}^2])$, and signal variance $s_a^2$: $k^a(\tilde{x}_i, \tilde{x}_j) = s_a^2 \exp\left(-\frac{1}{2}(\tilde{x}_i - \tilde{x}_j)^\top \Lambda_a^{-1}(\tilde{x}_i - \tilde{x}_j)\right)$.

### 3.3  Learning dynamics from noisy observations

The original PILCO algorithm ignored sensor noise when training each GP by assuming each observation $z_t$ to be the latent state $x_t$. However, this approximation breaks down under significant noise. More complex training schemes are required for each GP that correctly treat each training

datum $x_t$ as latent, yet noisily-observed as $z_t$. We resort to GP state space model methods, specifically the 'Direct method' [9, section 3.5]. The Direct method infers the marginal likelihood $p(z_{1:N})$ approximately using moment-matching in a single forward-pass. Doing so, it specifically exploits the time series structure that generated observations $z_{1:N}$. We use the Direct method to set the GP's training data $\{x_{1:N}, u_{1:N}\}$ and observation noise variance $\Sigma^\epsilon$ to the inducing point parameters and noise parameters that optimise the marginal likelihood. In this paper we use the superior Direct method to train GPs, both in our extended version of PILCO presented section 4, and in our implementation of the original PILCO algorithm for fair comparison in the experiments.

### 3.4 Prediction phase

In contrast to the execution phase, PILCO also *predicts* analytic distributions of state-trajectories (Algorithm 1, line 6) for policy evaluation. PILCO does this offline, between the online system executions. Predicted control is identical to executed control except each aforementioned quantity is instead now a random variable, distinguished with capitals: $X_t, Z_t, U_t, \tilde{X}_t$ and $X_{t+1}$, all approximated as jointly Gaussian. These variables interact both in execution and prediction according to Figure 1. To predict $X_{t+1}$ now that $\tilde{X}_t$ is uncertain PILCO uses the iterated law of expectation and variance:

$$p(X_{t+1}|\tilde{X}_t) = \mathcal{N}(\mu_{t+1}^x = \mathbb{E}_{\tilde{X}}[\mathbb{E}_f[f(\tilde{X}_t)]], \quad \Sigma_{t+1}^x = \mathbb{V}_{\tilde{X}}[\mathbb{E}_f[f(\tilde{X}_t)]] + \mathbb{E}_{\tilde{X}}[\mathbb{V}_f[f(\tilde{X}_t)]]). \quad (1)$$

After a one-step prediction from $X_0$ to $X_1$, PILCO repeats the process from $X_1$ to $X_2$, and up to $X_T$, resulting in a multi-step prediction whose joint we refer to as a distribution over state-trajectories.

## 4 Our method: PILCO extended with Bayesian filtering

Here we describe the novel aspects of our method. Our method uses the same high-level algorithm as PILCO (Algorithm 1). However, we modify (using PILCO's source code http://mlg.eng. cam.ac.uk/pilco/) two subroutines to extend PILCO from MDPs to a special-case of POMDPs (specifically where the partial observability has the form of additive Gaussian noise on the unobserved state $X$). First, we filter observations during system execution (Algorithm 1, line 4), detailed in Section 4.1. Second, we predict *belief*-trajectories instead of state-trajectories (line 6), detailed section 4.2. Filtering maintains a belief posterior of the latent system state. The belief is conditioned on, not just the most recent observation, but all previous observations (Figure 2). Such additional conditioning has the benefit of providing a less-noisy and more-informed input to the policy: the filtered belief-mean instead of the raw observation $z_t$. Our implementation continues PILCO's distinction between *executing* the system (resulting in a single real belief-trajectory) and *predicting* the system's responses (which in our case yields an analytic distribution of multiple possible future belief-trajectories). During the execution phase, the system reads specific observations $z_t$. Our method additionally maintains a belief state $b \sim \mathcal{N}(m, V)$ by filtering observations. This belief state $b$ can be treated as a random variable with a distribution parameterised by belief-mean $m$ and belief-certainty $V$ seen Figure 3. Note both $m$ and $V$ are functions of previous observations $z_{1:t}$. Now, during the (probabilistic) prediction phase, future observations are instead *random* variables (since they have not been observed yet), distinguished as $Z$. Since the belief parameters $m$ and $V$ are

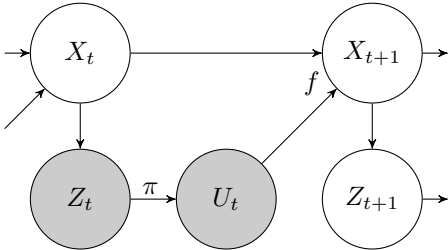

Figure 1: **The original (unfiltered) PILCO, as a probabilistic graphical model.** At each timestep, the latent system $X_t$ is observed noisily as $Z_t$ which is inputted directly into policy function $\pi$ to decide action $U_t$. Finally, the latent system will evolve to $X_{t+1}$, according to the unknown, nonlinear dynamics function $f$ of the previous state $X_t$ and action $U_t$.

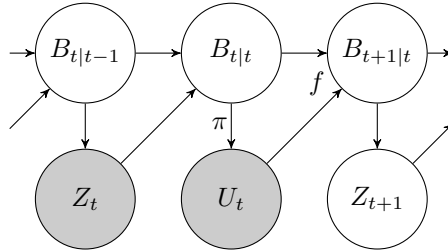

Figure 2: **Our method (PILCO extended with Bayesian filtering)**. Our prior belief $B_{t|t-1}$ (over latent system $X_t$), generates observation $Z_t$. The prior belief $B_{t|t-1}$ then combines with observation $Z_t$ resulting in posterior belief $B_{t|t}$ (the update step). Then, the mean posterior belief $\mathbb{E}[B_{t|t}]$ is inputted into policy function $\pi$ to decide action $U_t$. Finally, the next timestep's prior belief $B_{t+1|t}$ is predicted using dynamics model $f$ (the prediction step).

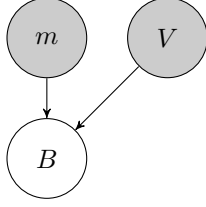

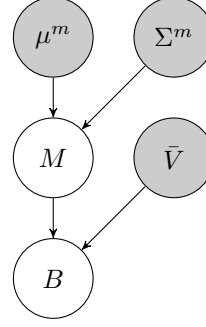

Figure 3: **Belief in execution phase**: a Gaussian random variable parameterised by mean $m$ and variance $V$.

Figure 4: **Belief in prediction phase**: a Gaussian random variable with random mean $M$ and non-random variance $\bar{V}$, where $M$ is itself a Gaussian random variable parameterised by mean $\mu^m$ and variance $\Sigma^m$.

functions of the now-random observations, the belief parameters must be random also, distinguished as $M$ and $V'$. Given the belief's distribution parameters are now random, the belief is *hierarchically-random*, denoted $B \sim \mathcal{N}(M, V')$ seen Figure 4. Our framework allows us to consider multiple possible future belief-states *analytically* during policy evaluation. Intuitively, our framework is an *analytical analogue* of POMDP policy evaluation using particle methods. In particle methods, each particle is associated with a distinct belief, due to each conditioning on independent samples of future observations. A particle distribution thus defines a distribution over beliefs. Our method is the analytical analogue of this particle distribution, and requires no sampling. By restricting our beliefs as (parametric) Gaussian, we can tractably encode a distribution over beliefs by a distribution over belief-parameters.

## 4.1 Execution phase with a filter

When an actual filter is applied, it starts with three pieces of information: $m_{t|t-1}$, $V_{t|t-1}$ and a noisy observation of the system $z_t$ (the dual subscript means belief of the latent physical state $x$ at time $t$ given all observations up until time $t-1$ inclusive). The filtering 'update step' combines prior belief $b_{t|t-1} = X_t|z_{1:t-1}, u_{1:t-1} \sim \mathcal{N}(m_{t|t-1}, V_{t|t-1})$ with observational likelihood $p(z_t) = \mathcal{N}(X_t, \Sigma^\epsilon)$ using Bayes rule to yield posterior belief $b_{t|t} = X_t|z_{1:t}, u_{1:t-1}$:

$$b_{t|t} \sim \mathcal{N}(m_{t|t}, V_{t|t}), \qquad m_{t|t} = W_m m_{t|t-1} + W_z z_t, \qquad V_{t|t} = W_m V_{t|t-1}, \qquad (2)$$

with weight matrices $W_m = \Sigma^\epsilon (V_{t|t-1} + \Sigma^\epsilon)^{-1}$ and $W_z = V_{t|t-1}(V_{t|t-1} + \Sigma^\epsilon)^{-1}$ computed from the standard result Gaussian conditioning. The policy $\pi$ instead uses updated belief-mean $m_{t|t}$ (smoother and better-informed than $z_t$) to decide the action: $u_t = \pi(m_{t|t}, \psi)$. Thus, the joint distribution over the updated (random) belief and the (non-random) action is

$$\tilde{b}_{t|t} \;\dot{=}\; \begin{bmatrix} b_{t|t} \\ u_t \end{bmatrix} \sim \mathcal{N}\left( \tilde{m}_{t|t} \dot{=} \begin{bmatrix} m_{t|t} \\ u_t \end{bmatrix}, \; \tilde{V}_{t|t} \dot{=} \begin{bmatrix} V_{t|t} & 0 \\ 0 & 0 \end{bmatrix} \right). \qquad (3)$$

Next, the filtering 'prediction step' computes the predictive-distribution of $b_{t+1|t} = p(x_{t+1}|z_{1:t}, u_{1:t})$ from the output of dynamics model $f$ given random input $\tilde{b}_{t|t}$. The distribution $f(\tilde{b}_{t|t})$ is non-Gaussian yet has analytically computable moments [5]. For tractability, we approximate $b_{t+1|t}$ as Gaussian-distributed using moment-matching:

$$b_{t+1|t} \sim \mathcal{N}(m_{t+1|t}, V_{t+1|t}), \qquad m_{t+1|t}^a = \mathbb{E}_{\tilde{b}_{t|t}}[f^a(\tilde{b}_{t|t})], \qquad V_{t+1|t}^{ab} = \mathbb{C}_{\tilde{b}_{t|t}}[f^a(\tilde{b}_{t|t}), f^b(\tilde{b}_{t|t})], \quad (4)$$

where $a$ and $b$ refer to the $a$'th and $b$'th dynamics output. Both $m_{t+1|t}^a$ and $V_{t+1|t}^{ab}$ are derived in Appendix D. The process then repeats using the predictive belief (4) as the prior belief in the following timestep. This completes the specification of the system in execution.

## 4.2 Prediction phase with a filter

During the prediction phase, we compute the probabilistic behaviour of the filtered system via an analytic distribution of belief states (Figure 4). We begin with a prior belief at time $t = 0$ before any observations are recorded (symbolised by '−1'), setting the prior Gaussian belief to have a distribution equal

to the *known* initial Gaussian state distribution: $B_{0|-1} \sim \mathcal{N}(M_{0|-1}, \bar{V}_{0|-1})$, where $M_{0|-1} \sim \mathcal{N}(\mu_0^x, 0)$ and $\bar{V}_{0|-1} = \Sigma_0^x$. Note the variance of $M_{0|-1}$ is zero, corresponding to a single prior belief at the beginning of the prediction phase. We probabilistically predict the *yet-unobserved* observation $Z_t$ using our belief distribution $B_{t|t-1}$ and the known additive Gaussian observation noise $\epsilon_t$ as per Figure 2. Since we restrict both the belief mean $M$ and observation $Z$ to being Gaussian random variables, we can express their joint distribution:

$$\begin{bmatrix} M_{t|t-1} \\ Z_t \end{bmatrix} \sim \mathcal{N}\left(\begin{bmatrix} \mu_{t|t-1}^m \\ \mu_{t|t-1}^m \end{bmatrix}, \begin{bmatrix} \Sigma_{t|t-1}^m & \Sigma_{t|t-1}^m \\ \Sigma_{t|t-1}^m & \Sigma_t^z \end{bmatrix}\right), \tag{5}$$

where $\Sigma_t^z = \Sigma_{t|t-1}^m + \bar{V}_{t|t-1} + \Sigma^\epsilon$.

The filtering 'update step' combines prior belief $B_{t|t-1}$ with observation $Z_t$ using the same logic as (2), the only difference being $Z_t$ is now random. Since the updated posterior belief mean $M_{t|t}$ is a (deterministic) function of *random* $Z_t$, then $M_{t|t}$ is necessarily random (with non-zero variance unlike $M_{0|-1}$). Their relationship, $M_{t|t} = W_m M_{t|t-1} + W_z Z_t$, results in the updated hierarchical belief posterior:

$$B_{t|t} \sim \mathcal{N}\left(M_{t|t}, \bar{V}_{t|t}\right), \text{ where } M_{t|t} \sim \mathcal{N}\left(\mu_{t|t}^m, \Sigma_{t|t}^m\right), \tag{6}$$

$$\mu_{t|t}^m = W_m \mu_{t|t-1}^m + W_z \mu_{t|t-1}^m = \mu_{t|t-1}^m, \tag{7}$$

$$\Sigma_{t|t}^m = W_m \Sigma_{t|t-1}^m W_m^\top + W_m \Sigma_{t|t-1}^m W_z^\top + W_z \Sigma_{t|t-1}^m W_m^\top + W_z \Sigma_t^z W_z^\top, \tag{8}$$

$$\bar{V}_{t|t} = W_m \bar{V}_{t|t-1}. \tag{9}$$

The policy now has a random input $M_{t|t}$, thus the control output must also be random (even though $\pi$ is a deterministic function): $U_t = \pi(M_{t|t}, \psi)$, which we implement by overloading the policy function: $(\mu_t^u, \Sigma_t^u, C_t^{mu}) = \pi(\mu_{t|t}^m, \Sigma_{t|t}^m, \psi)$, where $\mu_t^u$ is the output mean, $\Sigma_t^u$ the output variance and $C_t^{mu}$ input-output covariance with premultiplied inverse input variance, $C_t^{mu} \doteq (\Sigma_{t|t}^m)^{-1}\mathbb{C}_M[M_{t|t}, U_t]$. Making a moment-matched approximation yields a joint Gaussian:

$$\tilde{M}_{t|t} \doteq \begin{bmatrix} M_{t|t} \\ U_t \end{bmatrix} \sim \mathcal{N}\left(\mu_{t|t}^{\tilde{m}} \doteq \begin{bmatrix} \mu_{t|t}^m \\ \mu_t^u \end{bmatrix}, \Sigma_{t|t}^{\tilde{m}} \doteq \begin{bmatrix} \Sigma_{t|t}^m & \Sigma_{t|t}^m C_t^{mu} \\ (C_t^{mu})^\top \Sigma_{t|t}^m & \Sigma_t^u \end{bmatrix}\right). \tag{10}$$

Finally, we probabilistically predict the belief-mean $M_{t+1|t} \sim \mathcal{N}(\mu_{t+1|t}^m, \Sigma_{t+1|t}^m)$ and the expected belief-variance $\bar{V}_{t+1|t} = \mathbb{E}_{\tilde{M}_{t|t}}[V'_{t+1|t}]$. To do this we use a novel generalisation of Gaussian process moment matching with uncertain inputs by Candela et al. [1] generalised to hierarchically-uncertain inputs detailed in Appendix E. We have now discussed the one-step prediction of the filtered system, from $B_{t|t-1}$ to $B_{t+1|t}$. Using this process repeatedly, from initial belief $B_{0|-1}$ we one-step predict to $B_{1|0}$, then to $B_{2|1}$, up to $B_{T|T-1}$.

## 5    Experiments

We test our algorithm on the cartpole swing-up problem (shown in Appendix A), a benchmark for comparing controllers of nonlinear dynamical systems. We experiment using a physics simulator by solving the differential equations of the system. Each episode begins with the pendulum hanging downwards. The goal is then to swing the pendulum upright, thereafter continuing to balance it. The use a cart mass of $m_c = 0.5$kg. A zero-order hold controller applies horizontal forces to the cart within range $[-10, 10]$N. The policy is a linear combination of 100 radial basis functions. Friction resists the cart's motion with damping coefficient $b = 0.1$Ns/m. Connected to the cart is a pole of length $l = 0.2$m and mass $m_p = 0.5$kg located at its endpoint, which swings due to gravity's acceleration $g = 9.82$m/s$^2$. An inexpensive camera observes the system. Frame rates of \$10 webcams are typically 30Hz at maximum resolution, thus the time discretisation is $\Delta t = 1/30s$. The state $x$ comprises the cart position, pendulum angle, and their time derivatives $x = [x_c, \theta, \dot{x}_c, \dot{\theta}]^\top$. We both randomly-initialise the system and set the initial belief of the system according to $B_{0|-1} \sim \mathcal{N}(M_{0|-1}, V_{0|-1})$ where $M_{0|-1} \sim \delta([0, \pi, 0, 0]^\top)$ and $V_{0|-1}^{1/2} = \text{diag}([0.2\text{m}, 0.2\text{rad}, 0.2\text{m/s}, 0.2\text{rad/s}])$. The camera's noise standard deviation is: $(\Sigma^\epsilon)^{1/2} = \text{diag}([0.03\text{m}, 0.03\text{rad}, \frac{0.03}{\Delta t}\text{m/s}, \frac{0.03}{\Delta t}\text{rad/s}])$, noting $0.03\text{rad} \approx 1.7°$. We use the $\frac{0.03}{\Delta t}$ terms since using a camera we cannot observe velocities directly but can estimate them with finite differences. Each episode has a two second time horizon (60 timesteps). The cost function we impose is $1 - \exp\left(-\frac{1}{2}d^2/\sigma_c^2\right)$ where $\sigma_c = 0.25m$ and $d^2$ is the squared Euclidean distance between the pendulum's end point and its goal.

We compare four algorithms: 1) PILCO by Deisenroth and Rasmussen [5] as a baseline (unfiltered execution, and unfiltered full-prediction); 2) the method by Dallaire et al. [3] (filtered execution, and filtered MAP-prediction); 3) the method by Deisenroth and Peters [4] (filtered execution, and unfiltered full-prediction); and lastly 4) our method (filtered execution, and filtered full-prediction). For clear comparison we first control for data and dynamics models, where each algorithm has access to the exact same data and exact same dynamics model. The reason is to eliminate variance in performance caused by different algorithms choosing different actions. We generate a single dataset by running the baseline PILCO algorithm for 11 episodes (totalling 22 seconds of system interaction). The independent variables of our first experiment are 1) the method of system prediction and 2) the method of system execution. Each policy is then optimised from the same initialisation using their respective prediction methods, before comparing performances. Afterwards, we experiment allowing each algorithm to collect its own data, and also experiment with various noise level.

## 6 Results and analysis

### 6.1 Results using a common dataset

We now compare algorithm performance, both predictive (Figure 5) and empirical (Figure 6). First, we analyse predictive costs per timestep (Figure 5). Since predictions are probabilistic, the costs have distributions, with the exception of Dallaire et al. [3] which predicts MAP trajectories and therefore has deterministic cost. Even though we plot distributed costs, policies are optimised w.r.t. expected total cost only. Using the same dynamics, the different prediction methods optimise different policies (with the exception of Deisenroth and Rasmussen [5] and Deisenroth and Peters [4], whose prediction methods are identical). During the first 10 timesteps, we note identical performance with maximum cost due to the non-zero time required to physically swing the pendulum up near the goal. Performances thereafter diverge. Since we predict w.r.t. a filtering process, less noise is predicted to be injected into the policy, and the optimiser can thus afford higher gain parameters w.r.t. the pole at balance point. If we linearise our policy around the goal point, our policy has a gain of -81.7N/rad w.r.t. pendulum angle, a larger-magnitude than both Deisenroth method gains of -39.1N/rad (negative values refer to *left* forces in Figure 11). This higher gain is advantageous here, corresponding to a more reactive system which is more likely to catch a falling pendulum. Finally, we note Dallaire et al. [3] predict very high performance. Without balancing the costs across multiple possible trajectories, the method instead optimises a sequence of deterministic states to near perfection.

To compare the predictive results against the empirical, we used 100 executions of each algorithm (Figure 6). First, we notice a stark difference between predictive and executed performances from Dallaire et al. [3], due to neglecting model uncertainty, suffering model bias. In contrast, the other methods consider uncertainty and have relatively unbiased predictions, judging by the similarity between predictive-vs-empirical performances. Deisenroth's methods, which differ only in execution, illustrate that filtering during execution-only can be better than no filtering at all. However, the major benefit comes when the policy is evaluated from multi-step predictions of a filtered system. Opposed to Deisenroth and Peters [4], our method's predictions reflect reality closer because we both predict and execute system trajectories using closed loop filtering control.

To test statistical significance of empirical cost differences given 100 executions, we use a Wilcoxon rank-sum test at each time step. Excluding time steps ranging $t = [0, 29]$ (whose costs are similar), the minimum $z$-score over timesteps $t = [30, 60]$ that our method has superior average-cost than each other methods follows: Deisenroth 2011 $\min(z) = 4.99$, Dallaire 2009's $\min(z) = 8.08$, Deisenroth 2012's $\min(z) = 3.51$. Since the minimum $\min(z) = 3.51$, we have $p > 99.9\%$ certainty our method's average empirical cost is superior than each other method.

### 6.2 Results of full reinforcement learning task

In the previous experiment we used a common dataset to compare each algorithm, to isolate and focus on how well each algorithm makes *use* of data, rather than also considering the different ways each algorithm *collects* different data. Here, we remove the constraint of a common dataset, and test the full reinforcement learning task by allowing each algorithm to collect its own data over repeated trials of the cart-pole task. Each algorithm is allowed 15 trials (episodes), repeated 10 times with different random seeds. For a particular re-run experiment and episode number, an algorithm's predicted loss is unchanged when repeatedly computed, yet the empirical loss differs due to random initial states, observation noise, and process noise. We therefore average the empirical results over 100 random executions of the controller at each episode and seed.

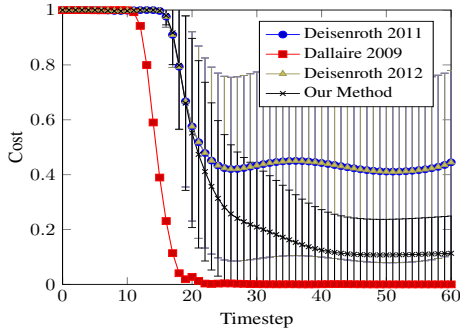

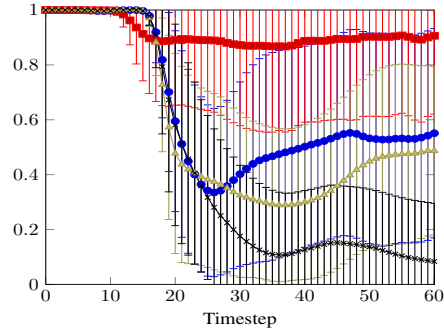

Figure 5: **Predictive cost per timestep.** The error bars show $\pm 1$ standard deviation. Each algorithm has access to the same data set (generated by baseline Deisenroth 2011) and dynamics model. Algorithms differ in their multi-step prediction methods (except Deisenroth's algorithms whose predictions overlap).

Figure 6: **Empirical cost per timestep**. We generate empirical cost distributions from 100 executions per algorithm. Error bars show $\pm 1$ standard deviation. The plot colours and shapes correspond to the legend in Figure 5.

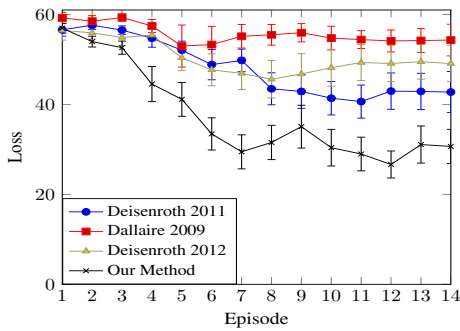

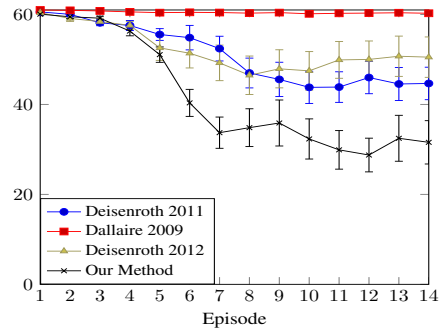

Figure 7: **Predictive loss per episode.** Error bars show $\pm 1$ standard error of the mean predicted loss given 10 repeats of each algorithm.

Figure 8: **Empirical loss per episode**. Error bars show $\pm 1$ standard error of the mean empirical loss given 10 repeats of each algorithm. In each repeat we computed the mean empirical loss using 100 independent executions of the controller.

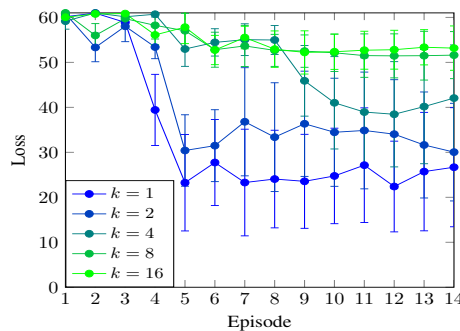

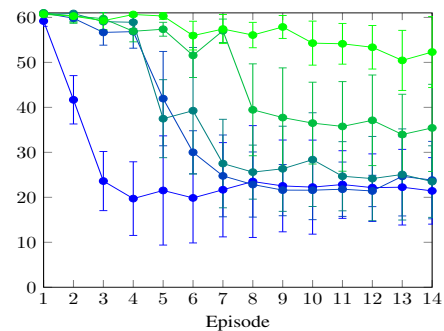

Figure 9: **Empirical loss of Deisenroth 2011 for various noise levels.** The error bars show $\pm 1$ standard deviation of the empirical loss distribution based on 100 repeats of the same learned controller, per noise level.

Figure 10: **Empirical loss of Filtered PILCO for various noise levels.** The error bars show $\pm 1$ standard deviation of the empirical loss distribution based on 100 repeats of the same learned controller, per noise level.

The predictive loss (cumulative cost) distributions of each algorithm are shown Figure 7. Perhaps the most striking difference between the full reinforcement learning predictions and those made with a controlled dataset (Figure 5) is that Dallaire does not predict it will perform well. The quality of the data collected by Dallaire within the first 15 episodes is not sufficient to predict good performance. Our Filtered PILCO method accurately predicts its own strong performance and additionally outperforms the competing algorithm seen in Figure 8. Of interest is how each algorithm performs equally poorly during the first four episodes, with Filtered PILCO's performance breaking away and learning the task well by the seventh trial. Such a learning rate was similar to the original PILCO experiment with the noise-free cartpole.

## 6.3 Results with various observation noises

Different observation noise levels were also tested, comparing PILCO (Figure 9) with Filtered PILCO (Figure 10). Both figures show a noise factors $k$, such that the observation noise is: $\sqrt{\Sigma^{\epsilon}} = k \times \text{diag}([0.01\text{m}, 0.01\text{rad}, \frac{0.01}{\Delta t}\text{m/s}, \frac{0.01}{\Delta t}\text{rad/s}])$. For reference, our previous experiments used a noise factor of $k = 3$. At low noise factor $k = 1$, both algorithms perform similarly-well, since observations are precise enough to control a system without a filter. As observations noise increases, the performance of unfiltered PILCO soon drops, whilst the Filtered PILCO can successfully control the system under higher noise levels (Figure 10).

## 6.4 Training time complexity

Training the GP dynamics model involved $N = 660$ data points, $M = 50$ inducing points under a sparse GP Fully Independent Training Conditional (FITC) [2], $P = 100$ policy RBF centroids, $D = 4$ state dimensions, $F = 1$ action dimensions, and $T = 60$ timestep horizon, with time complexity $\mathcal{O}(DNM^2)$. Policy optimisation (with 300 steps, each of which require trajectory prediction with gradients) is the most intense part: our method and both Deisenroth's methods scale $\mathcal{O}(M^2D^2(D+F)^2T + P^2D^2F^2T)$, whilst Dallaire's only scales $\mathcal{O}(MD(D+F)T + PDFT)$. Worst case we require $M = \mathcal{O}(\exp(D+F))$ inducing points to capture dynamics, the average case is unknown. Total training time was four hours to train the original PILCO method with an additional one hour to re-optimise the policy.

# 7 Conclusion and future work

In this paper, we extended the original PILCO algorithm [5] to filter observations, both during system execution and multi-step probabilistic prediction required for policy evaluation. The extended framework enables learning in a special case of partially-observed MDP environments (POMDPs) whilst retaining PILCO's data-efficiency property. We demonstrated successful application to a benchmark control problem, the noisily-observed cartpole swing-up. Our algorithm learned a good policy under significant observation noise in less than 30 seconds of system interaction. Importantly, our algorithm evaluates policies with predictions that are faithful to reality: we predict w.r.t. closed loop filtered control precisely because we execute closed loop filtered control. We showed experimentally that *faithful* and *probabilistic* predictions improved performance with respect to the baselines. For clear comparison we first constrained each algorithm to use the same dynamics dataset to demonstrate superior data-usage of our algorithm. Afterwards we relaxed this constraint, and showed our algorithm was able to learn from fewer data.

Several more challenges remain for future work. Firstly the assumption of zero variance of the belief-variance could be relaxed. A relaxation allows distributed trajectories to more accurately consider belief states having various degrees of certainty (belief-variance). For example, system trajectories have larger belief-variance when passing though data-sparse regions of state-space, and smaller belief-variance in data-dense regions. Secondly, the policy could be a function of the full belief distribution (mean and variance) rather than just the mean. Such flexibility could help the policy make more 'cautious' actions when more uncertain about the state. A third challenge is handling non-Gaussian noise and unobserved state variables. For example, in real-life scenarios using a camera sensor for self-driving, observations are occasionally fully or partially occluded, or limited by weather conditions, where such occlusions and limitations change, opposed to assuming a fixed Gaussian addition noise. Lastly, experiments with a real robot would be important to show the usefulness in practice.

## Footnotes

[1] The original PILCO [5] instead uses a zero mean function, and instead predicts relative changes in state.

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
