[Supplementary Material]

# A The cartpole system

We test our algorithm on the cartpole swing-up problem (Figure sec:app-cartpole). The cartpole's motion is described with the differential equation:

$$\dot{x}^\top = \left[\dot{x}_c, \dot{\theta}, \frac{-2m_p l \dot{\theta}^2 s + 3m_p gsc + 4u - 4b\dot{x}_c}{4(m_c + m_p) - 3m_p c^2}, \frac{-3m_p l \dot{\theta}^2 sc + 6(m_c + m_p)gs + 6(u - b\dot{x}_c)c}{4l(m_c + m_p) - 3m_p lc^2}\right], \quad (11)$$

using shorthand $s = \sin\theta$ and $c = \cos\theta$. The cost function we impose is $1 - \exp\left(-\frac{1}{2}d^2/\sigma_c^2\right)$ where $\sigma_c = 0.25m$ and $d^2$ is the squared Euclidean distance between the pendulum's end point $(x_p, y_p)$ and its goal $(0, l)$. I.e. $d^2 = x_p^2 + (l - y_p)^2 = (x_c - l\sin\theta)^2 + (l - l\cos\theta)^2$.

Figure 11: **The cartpole swing-up task.** A pendulum of length $l$ is attached to a cart by a frictionless pivot. The cart has mass $m_c$ and position $x_c$. The pendulum's endpoint has mass $m_p$ and position $(x_p, y_p)$, with angle $\theta$ from vertical. The cart begins at position $x_c = 0$ and pendulum hanging down: $\theta = \pi$. The goal is to accelerate the cart by applying horizontal force $u_t$ at each timestep $t$ to invert then stabilise the pendulum's endpoint at the goal (black cross), i.e. to maintain $x_c = 0$ and $\theta = 0$.

# B Gradients for policy improvement

Let $\text{vec}(\cdot)$ be the 'unwrap operator' that reshapes matrices columnwise into vectors. We define a Markov filtered-system from the belief's parameters: $S_t = [M_{t|t-1}^\top, \text{vec}(V_{t|t-1})^\top]^\top$. To predict system evolution, the state distribution is defined:

$$p(S_t) \sim \mathcal{N}\left(\mu_t^s = \begin{bmatrix} \mu_{t|t-1}^m \\ \text{vec}(V_{t+1|t}) \end{bmatrix}, \quad \Sigma_t^s = \begin{bmatrix} \Sigma_{t|t-1}^m & 0 \\ 0 & 0 \end{bmatrix}\right). \quad (12)$$

To compute policy gradient $\mathrm{d}J/\mathrm{d}\psi$ we first require $\mathrm{d}\mathcal{E}_t/\mathrm{d}\psi$:

$$\frac{\mathrm{d}\mathcal{E}_t}{\mathrm{d}\theta} = \frac{\mathrm{d}\mathcal{E}_t}{\mathrm{d}p(S_t)}\frac{\mathrm{d}p(S_t)}{\mathrm{d}\theta}$$

$$= \frac{\mathrm{d}\mathcal{E}_t}{\partial\mu_t^s}\frac{\partial\mu_t^s}{\mathrm{d}\theta} + \frac{\mathrm{d}\mathcal{E}_t}{\partial\Sigma_t^s}\frac{\partial\Sigma_t^s}{\mathrm{d}\theta}, \qquad \text{and} \qquad (13)$$

$$\frac{\mathrm{d}p(S_{t+1})}{\mathrm{d}\theta} = \frac{\mathrm{d}p(S_{t+1})}{\mathrm{d}p(S_t)}\frac{\mathrm{d}p(S_t)}{\mathrm{d}\theta} + \frac{\partial p(S_{t+1})}{\partial\theta}. \qquad (14)$$

Application of the chain rule backwards from the state distribution at the horizon $S_T$, to $S_t$ at arbitrary time $t$, is analogous to that detailed in PILCO [5], where we use $S_t$, $\mu_t^s$ and $\Sigma_t^s$ in the place of $x_t$, $\mu_t$ and $\Sigma_t$ respectively.

## B.1 Policy evaluation and improvement

To evaluate the policy $\pi$ (or more specifically, the policy parameters $\psi$), PILCO computes the loss $J(\psi)$ by applying a cost function to the marginal state distribution at each timestep (see Algorithm 1, line 7). After policy evaluation, PILCO optimises the policy using the analytic gradients of $J$. A BFGS optimisation method searches for the set of policy parameters $\psi$ that minimise the total cost $J(\psi)$ using gradients information $\mathrm{d}J/\mathrm{d}\psi$ (Algorithm 1, line 8). To compute $\mathrm{d}J/\mathrm{d}\psi$ we require derivatives $\mathrm{d}\mathcal{E}_t/\mathrm{d}\psi$ at each time $t$ to chain together, detailed in [5].

## B.2 Policy evaluation and improvement with a filter

To evaluate a policy we again apply the loss function $J$ (Algorithm 1, line 7) to the multi-step prediction (section 4.2). The policy is again optimised using the analytic gradients of $J$. Since $J$ now is a function of beliefs, we additionally consider the gradients of $B_{t|t-1}$ w.r.t. $\psi$. As the belief is distributed by $B_{t|t-1} \sim \mathcal{N}(M_{t|t-1}, V_{t|t-1}) \sim \mathcal{N}(\mathcal{N}(\mu_{t|t-1}^m, \Sigma_{t|t-1}^m), V_{t|t-1})$, we use partial derivatives of $\mu_{t|t-1}^m$, $\Sigma_{t|t-1}^m$ and $V_{t|t-1}$ w.r.t. each other and w.r.t $\psi$.

# C Identities for Gaussian process prediction with hierarchical uncertain inputs

The two functions

$$q(x, x', \Lambda, V) \triangleq |\Lambda^{-1}V + I|^{-1/2} \exp\left(-\tfrac{1}{2}(x - x')[\Lambda + V]^{-1}(x - x')\right),$$

$$
\begin{aligned}
Q(x, x', \Lambda_a, \Lambda_b, V, \mu, \Sigma) &\triangleq c_1 \exp\left(-\tfrac{1}{2}(x - x')^\top[\Lambda_a + \Lambda_b + 2V]^{-1}(x - x')\right) \\
&\quad \times \exp\left(-\tfrac{1}{2}(z - \mu)^\top\left[\left((\Lambda_a + V)^{-1} + (\Lambda_b + V)^{-1}\right)^{-1} + \Sigma\right]^{-1}(z - \mu)\right), \\
&= c_2\, q(x, \mu, \Lambda_a, V)\, q(\mu, x'\Lambda_b, V) \\
&\quad \times \exp\left(\tfrac{1}{2}\mathbf{r}^\top\left[(\Lambda_a + V)^{-1} + (\Lambda_b + V)^{-1} + \Sigma^{-1}\right]^{-1}\mathbf{r}\right),
\end{aligned}
$$

$$\text{where}\quad
\begin{cases}
z &= (\Lambda_b + V)(\Lambda_a + \Lambda_b + 2V)^{-1}x + (\Lambda_a + V)(\Lambda_a + \Lambda_b + 2V)^{-1}x' \\
\mathbf{r} &= (\Lambda_a + V)^{-1}(x - \mu) + (\Lambda_b + V)^{-1}(x' - \mu) \\
c_1 &= \left|(\Lambda_a + V)(\Lambda_b + V) + (\Lambda_a + \Lambda_b + 2V)\Sigma\right|^{-1/2}\left|\Lambda_a\Lambda_b\right|^{1/2} \\
c_2 &= \left|\left((\Lambda_a + V)^{-1} + (\Lambda_b + V)^{-1}\right)\Sigma + I\right|^{-1/2},
\end{cases}
\tag{15}
$$

have the following Gaussian integrals

$$\int q(x, t, \Lambda, V)\mathcal{N}(t|\mu, \Sigma)dt = q(x, \mu, \Lambda, \Sigma + V),$$

$$\int q(x, t, \Lambda_a, V)\, q(t, x', \Lambda_b, V)\, \mathcal{N}(t|\mu, \Sigma)dt = Q(x, x', \Lambda_a, \Lambda_b, V, \mu, \Sigma), \tag{16}$$

$$\int Q(x, x', \Lambda_a, \Lambda_b, 0, \mu, V)\mathcal{N}(\mu|\mathbf{m}, \Sigma)d\mu = Q(x, x', \Lambda_a, \Lambda_b, 0, \mathbf{m}, \Sigma + V).$$

We want to model data with $E$ output coordinates, and use separate combinations of linear models and GPs to make predictions, $a = 1, \ldots, E$:

$$f_a(x^*) = f_a^* \sim \mathcal{N}\left(\theta_a^\top x^* + k_a(x^*, \mathbf{x})\beta_a,\ k_a(x^*, x^*) - k_a(x^*, \mathbf{x})(K_a + \Sigma_\varepsilon^a)^{-1}k_a(\mathbf{x}, x^*)\right),$$

where the $E$ squared exponential covariance functions are

$$k_a(x, x') = s_a^2 q(x, x', \Lambda_a, 0), \quad \text{where}\ a = 1, \ldots, E, \tag{17}$$

and $s_a^2$ are the signal variances and $\Lambda_a$ is a diagonal matrix of squared length scales for GP number $a$. The noise variances are $\Sigma_\varepsilon^a$. The inputs are $\mathbf{x}$ and the outputs $y_a$ and we define $\beta_a = (K_a + \Sigma_\varepsilon^a)^{-1}(y_a - \theta_a^\top \mathbf{x})$, where $K_a$ is the Gram matrix.

## C.1 Derivatives

For symmetric $\Lambda$ and $V$ and $\Sigma$:

$$
\begin{aligned}
\frac{\partial \ln q(x, x', \Lambda, V)}{\partial x} &= -(\Lambda + V)^{-1}(x - x') = -(\Lambda^{-1}V + I)^{-1}\Lambda^{-1}(x - x') \\
\frac{\partial \ln q(x, x', \Lambda, V)}{\partial x'} &= (\Lambda + V)^{-1}(x - x') \\
\frac{\partial \ln q(x, x', \Lambda, V)}{\partial V} &= -\frac{1}{2}(\Lambda + V)^{-1} + \frac{1}{2}(\Lambda + V)^{-1}(x - x')(x - x')^\top(\Lambda + V)^{-1}
\end{aligned}
\tag{18}
$$

Let $L = (\Lambda_a + V)^{-1} + (\Lambda_b + V)^{-1}$, $R = \Sigma L + I$, $Y = R^{-1}\Sigma = \left[L + \Sigma^{-1}\right]^{-1}$, $T : X \to XX^\top$:

$$\partial Q(x, x', \Lambda_a, \Lambda_b, V, \mu, \Sigma) = Q \circ \partial\left(\ln c_2 + \ln q(x, \mu, \Lambda_a, V) + \ln q(\mu, x'\Lambda_b, V) + \frac{1}{2}y^\top Y y\right)$$

$$\frac{1}{2}\frac{\partial\, y^\top Y y}{\partial \mu} = y^\top Y \frac{\partial y}{\partial \mu} = -y^\top Y L$$

$$\frac{\partial \ln c_2}{\partial \Sigma} = -\frac{1}{2}\frac{\partial \ln |L\Sigma + I|}{\partial \Sigma} = -\frac{1}{2}L^\top (L\Sigma + I)^{-\top} = -\frac{1}{2}LR^{-1}$$

$$\frac{\partial\, y^\top Y y}{\partial \Sigma} = \Sigma^{-\top}Y^\top yy^\top Y^\top \Sigma^{-\top} = T(R^{-\top}y)$$

$$\frac{\partial \ln c_2}{\partial V} = -\frac{1}{2}\frac{\partial \ln |L\Sigma + I|}{\partial V} = -\frac{1}{2}\frac{\partial \ln \left|\sum_i \left[(\Lambda_i + V)^{-1}\right]\Sigma + I\right|}{\partial V}$$

$$= \frac{1}{2}\sum_i \left[(\Lambda_i + V)^{-\top}\left(\sum_j \left[(\Lambda_j + V)^{-1}\right]\Sigma + I\right)^{-\top}\Sigma^\top (\Lambda_i + V)^{-\top}\right]$$

$$= \frac{1}{2}\sum_i \left[(\Lambda_i + V)^{-1}Y(\Lambda_i + V)^{-1}\right]$$

$$\frac{\partial\, y^\top Y y}{\partial V} = y^\top \frac{\partial Y}{\partial V}y + \frac{\partial y^\top}{\partial V}Yy + y^\top Y \frac{\partial y}{\partial V} = \sum_i \left[(\Lambda_i + V)^{-1}Y^\top yy^\top Y^\top (\Lambda_i + V)^{-1}\right]$$

$$- \sum_i \left[(\Lambda_i + V)^{-1}(x_{n_i} - \mu)(Yy)^\top (\Lambda_i + V)^{-1}\right]$$

$$- \sum_i \left[(\Lambda_i + V)^{-1}(y^\top Y)^\top (x_{n_i} - \mu)^\top (\Lambda_i + V)^{-1}\right]$$

$$= \sum_i \left[T\left((\Lambda_i + V)^{-1}(Yy - (x_{n_i} - \mu))\right) - T\left((\Lambda_i + V)^{-1}(x_{n_i} - \mu)\right)\right]$$

$$(19)$$

## D  Dynamics predictions in execution phase

Here we specify the predictive distribution $p(b_{t+1|t})$, whose moments are equal to the moments from dynamics model output $f$ with uncertain input $\tilde{b}_{t|t} \sim \mathcal{N}(\tilde{m}_{t|t}, \tilde{V}_{t|t})$ similar to Deisenroth and Rasmussen [5] which was based on work by Candela et al. [1]. Consider making predictions from $a = 1, \ldots, E$ GPs at $\tilde{b}_{t|t}$ with specification $\tilde{b}_{t|t} \sim \mathcal{N}(\tilde{m}_{t|t}, \tilde{V}_{t|t})$. We have the following expressions for the predictive mean, variances and input-output covariances using the law of iterated expectations and variances:

$$
\begin{aligned}
b_{t+1|t} &\sim \mathcal{N}(m_{t+1|t}, V_{t+1|t}), &(20)\\
m_{t+1|t}^a &= \mathbb{E}_{\tilde{b}_{t|t}}[f^a(\tilde{b}_{t|t})]\\
&= \int \big(s_a^2 \beta_a^\top q(x_i, \tilde{b}_{t|t}, \Lambda_a, 0) + \phi_a^\top \tilde{b}_{t|t}\big) \mathcal{N}(\tilde{b}_{t|t}; \tilde{m}_{t|t}, \tilde{V}_{t|t}) d\tilde{b}_{t|t}\\
&= s_a^2 \beta_a^\top q^a + \phi_a^\top \tilde{m}_{t|t}, &(21)\\
C_a &\doteq \tilde{V}_{t|t}^{-1} \mathbb{C}_{\tilde{b}_{t|t}}[\tilde{b}_{t|t}, f^a(\tilde{b}_{t|t}) - \phi_a^\top \tilde{b}_{t|t}],\\
&= \tilde{V}_{t|t}^{-1} \int (\tilde{b}_{t|t} - \tilde{m}_{t|t}) s_a^2 \beta_a^\top q(\mathrm{x}, \tilde{b}_{t|t}, \Lambda_a, 0) \mathcal{N}(\tilde{b}_{t|t}; \tilde{m}_{t|t}, \tilde{V}_{t|t}) d\tilde{b}_{t|t}\\
&= s_a^2 (\Lambda_a + \tilde{V}_{t|t})^{-1} (\mathrm{x} - \tilde{m}_{t|t}) \beta_a q^a, &(22)\\
V_{t+1|t}^{ab} &= \mathbb{C}_{\tilde{b}_{t|t}} \Big[f^a(\tilde{b}_{t|t}), \; f^b(\tilde{b}_{t|t})\Big]\\
&= \mathbb{C}_{\tilde{b}_{t|t}} \Big[\mathbb{E}_f[f^a(\tilde{b}_{t|t}), \mathbb{E}_f[f^b(\tilde{b}_{t|t})]\Big] + \mathbb{E}_{\tilde{b}_{t|t}} \Big[\mathbb{C}_f[f^a(\tilde{b}_{t|t}), \; f^b(\tilde{b}_{t|t})]\Big]\\
&= \mathbb{C}_{\tilde{b}_{t|t}} \Big[s_a^2 \beta_a^\top q(\mathrm{x}, \tilde{b}_{t|t}, \Lambda_a, 0) + \phi_a^\top \tilde{b}_{t|t}, \;\; s_b^2 \beta_b^\top q(\mathrm{x}, \tilde{b}_{t|t}, \Lambda_b, 0) + \phi_b^\top \tilde{b}_{t|t}\Big] +\\
&\quad \delta_{ab} \mathbb{E}[s_a^2 - k_a(\tilde{b}_{t|t}, \mathrm{x})(K_a + \Sigma_\varepsilon^a)^{-1} k_a(\mathrm{x}, \tilde{b}_{t|t})]\\
&= s_a^2 s_b^2 \big[\beta_a^\top (Q^{ab} - q^a q^{b\top}) \beta_b +\\
&\quad \delta_{ab} \big(s_a^{-2} - \mathrm{tr}((K_a + \Sigma_\varepsilon^a)^{-1} Q^{aa})\big)\big] + C_a^\top \tilde{V}_{t|t} \phi_b + \phi_a^\top \tilde{V}_{t|t} C_b + \phi_a^\top \tilde{V}_{t|t} \phi_b, &(23)
\end{aligned}
$$

where

$$
\begin{aligned}
q_i^a &= q\big(\mathrm{x}_i, \tilde{m}_{t|t}, \Lambda_a, \tilde{V}_{t|t}\big),\\
Q_{ij}^{ab} &= Q\big(\mathrm{x}_i, \mathrm{x}_j, \Lambda_a, \Lambda_b, 0, \tilde{m}_{t|t}, \tilde{V}_{t|t}\big),\\
\beta_a &= (K_a + \Sigma^{\epsilon,a})^{-1} (y_a - \phi_a^\top \mathrm{x}),
\end{aligned}
$$

and training inputs are $\mathrm{x}$, outputs are $y_a$ (determined by the 'Direct method'), $K_a$ is a Gram matrix.

## E  Dynamics predictions in prediction phase

Here we describe the prediction formulae for the random belief state in the prediction phase. We again note, during execution, our belief distribution is specified by certain parameters, $b_{t|t} \sim \mathcal{N}(m_{t|t}, \tilde{V}_{t|t})$. By contrast, during the prediction phase, our belief distribution is specified by an uncertain belief-mean and certain belief-variance: $B_{t|t} \sim \mathcal{N}(M_{t|t}, \bar{V}_{t|t}) \sim \mathcal{N}(\mathcal{N}(\mu_{t|t}^m, \Sigma_{t|t}^m), \bar{V}_{t|t})$, where we assumed a delta distribution on $\bar{V}_{t|t}$ for mathematical simplicity, i.e. $\mathrm{vec}(\bar{V}_{t|t}) \sim \mathcal{N}(\mathrm{vec}(\bar{V}_{t|t}), 0)$. Therefore we conduct GP prediction given hierarchically-uncertain inputs, outlining each output moment below. For instance, consider making predictions from $a = 1, \ldots, E$ GPs at $B_{t|t}$ with *hierarchical* specification

$$
B_{t|t} \sim \mathcal{N}(M_{t|t}, \bar{V}_{t|t}), \;\; \text{and} \;\; M_{t|t} \sim \mathcal{N}(\mu_{t|t}^m, \Sigma_{t|t}^m), \tag{24}
$$

or equivalently the joint

$$
\begin{bmatrix} B_{t|t} \\ M_{t|t} \end{bmatrix} \sim \mathcal{N}\left( \begin{bmatrix} \mu_{t|t}^m \\ \mu_{t|t}^m \end{bmatrix}, \begin{bmatrix} \Sigma_{t|t}^m + \bar{V}_{t|t} & \Sigma_{t|t}^m \\ \Sigma_{t|t}^m & \Sigma_{t|t}^m \end{bmatrix} \right). \tag{25}
$$

**Mean of the Belief-Mean:** dynamics prediction uses input $\tilde{M}_{t|t} \sim \mathcal{N}(\mu_{t|t}^{\tilde{m}}, \Sigma_{t|t}^{\tilde{m}})$, which is jointly distributed according to (10). Using the belief-mean $m_{t+1|t}^a$ definition (21),

$$
\begin{aligned}
\mu_{t+1|t}^{m,a} &= \mathbb{E}_{\tilde{M}_{t|t}}[M_{t+1|t}^a] \\
&= \int M_{t+1|t}^a \mathcal{N}(\tilde{M}_{t|t}; \mu_{t|t}^{\tilde{m}}, \Sigma_{t|t}^{\tilde{m}}) \mathrm{d}\tilde{M}_{t|t}, \\
&= s_a^2 \beta_a^\top \int q(\mathrm{x}, \tilde{M}_{t|t}, \Lambda_a, \tilde{V}_{t|t}) \mathcal{N}(\tilde{M}_{t|t}; \mu_{t|t}^{\tilde{m}}, \Sigma_{t|t}^{\tilde{m}}) d\tilde{M}_{t|t} + \phi_a^\top \mu_{t|t}^{\tilde{m}} \\
&= s_a^2 \beta_a^\top \hat{q}^a + \phi_a^\top \mu_{t|t}^{\tilde{m}}, && (26) \\
\hat{q}_i^a &= q\Big(x_i, \mu_{t|t}^{\tilde{m}}, \Lambda_a, \Sigma_{t|t}^{\tilde{m}} + \tilde{V}_{t|t}\Big). && (27)
\end{aligned}
$$

**Input-Output Covariance:** the expected input-output covariance belief term (22) (equivalent to the input-output covariance of the belief-mean) is:

$$
\begin{aligned}
\hat{C}_a &\doteq \tilde{V}_{t|t}^{-1} \mathbb{E}_{\tilde{M}_{t|t}}[\mathbb{C}_{B_{t|t}}[\tilde{B}_{t|t}, f(\tilde{B}_{t|t}) - \phi_a^\top \tilde{M}_{t|t}]], \quad \text{and similarly defined} \\
&\doteq (\Sigma_{t|t}^{\tilde{m}})^{-1} \mathbb{C}_{\tilde{M}_{t|t}}[\tilde{M}_{t|t}, \mathbb{E}_{B_{t|t}}[f(\tilde{B}_{t|t}) - \phi_a^\top \tilde{M}_{t|t}]], \\
&= (\Sigma_{t|t}^{\tilde{m}})^{-1} \int (\tilde{M}_{t|t} - \mu_{t|t}^{\tilde{m}}) \mathbb{E}_{B_{t|t}}[f(\tilde{B}_{t|t})] \mathcal{N}(\tilde{M}_{t|t}; \mu_{t|t}^{\tilde{m}}, \Sigma_{t|t}^{\tilde{m}}) d\tilde{M}_{t|t} \\
&= (\Sigma_{t|t}^{\tilde{m}})^{-1} \int (\tilde{M}_{t|t} - \mu_{t|t}^{\tilde{m}}) \big(s_a^2 \beta_a^\top q(x_i, \tilde{M}_{t|t}, \Lambda_a, \tilde{V}_{t|t})\big) \mathcal{N}(\tilde{M}_{t|t}; \mu_{t|t}^{\tilde{m}}, \Sigma_{t|t}^{\tilde{m}}) d\tilde{M}_{t|t} \\
&= s_a^2 (\Lambda_a + \Sigma_{t|t}^{\tilde{m}} + \tilde{V}_{t|t})^{-1} (\mathrm{x} - \mu_{t|t}^{\tilde{m}}) \beta_a \hat{q}_i^a. && (28)
\end{aligned}
$$

**Variance of the Belief-Mean:** the variance of randomised belief-mean (Eq 21) is:

$$
\begin{aligned}
\Sigma_{t+1|t}^{m,ab} &= \mathbb{C}_{\tilde{M}_{t|t}}[M_{t+1|t}^a, M_{t+1|t}^b], \\
&= \int M_{t+1|t}^a M_{t+1|t}^b \mathcal{N}(\tilde{M}_{t|t}|\mu_{t|t}^{\tilde{m}}, \Sigma_{t|t}^{\tilde{m}}) \mathrm{d}\tilde{M}_{t|t} - \mu_{m_{t+1|t}}^a \mu_{m_{t+1|t}}^b, \\
&= s_a^2 s_b^2 \beta_a^\top (\hat{Q}^{ab} - \hat{q}^a \hat{q}^{b\top}) \beta_b + \hat{C}_a^\top \Sigma_{t|t}^{\tilde{m}} \phi_b + \phi_a^\top \Sigma_{t|t}^{\tilde{m}} \hat{C}_b + \phi_a^\top \Sigma_{t|t}^{\tilde{m}} \phi_b, && (29) \\
\hat{Q}_{ij}^{ab} &= Q(\mathrm{x}_i, \mathrm{x}_j, \Lambda_a, \Lambda_b, \tilde{V}_{t|t}, \mu_{t|t}^{\tilde{m}}, \Sigma_{t|t}^{\tilde{m}}). && (30)
\end{aligned}
$$

**Mean of the Belief-Variance:** using the belief-variance $V_{t+1|t}^{ab}$ definition (23),

$$
\begin{aligned}
\bar{V}_{t+1|t}^{ab} &= \mathbb{E}_{\tilde{M}_{t|t}}[V_{t+1|t}^{ab}] \\
&= \int V_{t+1|t}^{ab} \mathcal{N}(\tilde{M}_{t|t}|\mu_{t|t}^{\tilde{m}}, \Sigma_{t|t}^{\tilde{m}}) \mathrm{d}\tilde{M}_{t|t} \\
&= s_a^2 s_b^2 \big[ \beta_a^\top (\tilde{Q}^{ab} - \hat{Q}^{ab}) \beta_b + \delta_{ab} \big(s_a^{-2} - \mathrm{tr}((K_a + \Sigma_\varepsilon^a)^{-1} \tilde{Q}^{aa})\big) \big] \\
&\quad + \hat{C}_a^\top \tilde{V}_{t|t} \phi_b + \phi_a^\top \tilde{V}_{t|t} \hat{C}_b + \phi_a^\top \tilde{V}_{t|t} \phi_b, && (31) \\
\tilde{Q}_{ij}^{ab} &= Q(\mathrm{x}_i, \mathrm{x}_j, \Lambda_a, \Lambda_b, 0, \mu_{t|t}^{\tilde{m}}, \Sigma_{t|t}^{\tilde{m}} + \tilde{V}_{t|t}). && (32)
\end{aligned}
$$