[Reviews · NeurIPS 2017]

Reviewer 1



This paper describes an extension to the PILCO algorithm (Probabilistic Inference and Learning for COntrol, a data-efficient reinforcement algorithm). The proposed algorithm applies a measurement filtering algorithm during the actual experiment and explicitly takes this measurement filtering algorithm into account during the policy learning step, which uses data from the experiment. This is an important practical extension addressing the fact that measurements are often very noisy. My intuitive explanation for this approach is that the proposed approach makes the overall feedback system more "repeatable" (noise is mostly filtered out) and therefore learning is faster (given that the filtering is effective, see last sentence of the conclusion). The paper presents detailed mathematical derivations and strong simulation results that highlight the properties of the proposed algorithm. Comments/Suggestions: - A discussion of the limitations of the work would be useful. It is only briefly mentioned in the Future Work section. E.g., how do you handle unobserved states? I think in practice this is a real limitation as we often do not know some of the underlying dependencies (e.g., the behavior of a self-driving car depends on weather conditions, road conditions etc). - The paper needs more attention to detail. There are various typos and inconsistencies in terms of spelling throughout the paper. Reference style within the paper is inconsistent (using author names and not using it). Examples: L19 account FOR (instead of "of"), L6/L29 observation space is spelled with and without hyphen, L68 into THE state, L68 reducing the learning task TO, L71 The original PILCO FRAMEWORK does, L92 over AN, L103 policy IS initialised, etc etc - I would appreciate if you provide/emphasize more intuitive explanations regarding the practical implications/interpretations of the algorithms (similar to my intuitive explanation above?) - Real experiments would be important to show the usefulness in practice. Should be part of future work?! - References need to be cleaned up. The reference style is not consistent. Refs. [10] and [11] are identical.

Reviewer 2



SUMMARY In this paper, the authors propose to learn and plan with a model that represents uncertainty in a specific case of partial observability - that of noisy observations. Their method the analogue of a known model based method - PILCO - for this noisy observation case. It handles the noise by assuming a joint Gaussian distribution over the latent states and observations at each time step when making predictions (at execution time, observations are known and a filtered latent state is used). On the kind of dynamical system that these method are commonly evaluated on - the cart-pole swing-up in this case - the proposed methods seems to be better than relevant baselines at dealing with noise. TECHNICAL QUALITY The proposed algorithm is supported by a derivation of the update equation and experimental validation on the cart-pole swing-up task. The derivation seems technically solid. I just wondered why in (5) the covariance between M_t|t-1 and Z_t is 0, since in Fig. 2 it seems that Z_t is a noisy version of B_t|t-1 of which M_t|t-1 is the mean. Although the experiments only consider a single system, the results seem quite convincing. The empirical loss per episode is significantly lower, and predictions of the cost per time step are much better than for the compared methods. I just wondered what is meant by the "empirical loss distribution", and whether the 100 independent runs that the error bars are based on are 100 evaluations of the same learned controller; or 100 independently trained controllers. In Figs. 7 and 8 I would be interested to see the n=0 case (or, if that causes numerical problems, n=0.01 or similar). In terms of related work, the authors mainly restrict themselves to discussing Bayesian model learning and Bayesian RL techniques. It might be interesting to also discuss other styles of policy search methods for the partially observable setting. It might be insightful to describe what about the current method limits us to the noisy observation case. If we observed positions but not velocities, coulnd't we filter that, too? NOVELTY I haven't seen earlier extensions of PILCO-type methods to the partially observable case. So I would consider the method quite novel. SIGNIFICANCE AND RELEVANCE Noisy dynamical systems occur in many (potential) application domain of reinforcement learning (robotics, smart grid, autonomous driving), so, this is a relevant research topic for the RL community. On the investigated system, the improvement relative to the state of the art seems quite significant, however, the method is only evaluated on a single system. CLARITY The paper is mostly well-written. However, in some part the clarity could be improved. For example, the paper does not mention explicitly they assume the reward function to be known. The policy improvement step (8 in the algorithm) is not explained at all. For readers already familiar with PILCO this is quite obvious but it would be good to give a short introduction from the others. The technical section in 4.1 - 4.2 is very dense. I understand space is rather limited but it could be good to explain this part in more detail in the appendix. For example, the paper states V=E[V'] is assumed fixed, but doesn't state how this fixed value is computed or set. Figures 1 and 2 are presented as analogues, yet the latent state X is not shown in the right image. It would be easier to understand the relationship if the underlying latent process was shown in both images. There are quite many grammar issues. I'll point some out under minor comments. MINOR COMMENTS Grammar - please rephrase sentences in lines 7, 52, 86-87, 134,209,250, 283, 299-300 Contractions and abbreviations - please write out (doesn't, etc.) Some sentence start with note but don't seem grammatical. I suggest writing "Note that, ..." or similar. Line 55: Greater -> Larger Line 65: Seems like the author name shouldn't appear here Line 99: Less -> fewer line 112: "Overfitting leads to bias". Overcapacity generally causes a high variance, not necessarily bias. Or is the bias here a result of optimizing the policy on the erroneous model? (similar in line 256) line 118: I guess k(x_i, x_j) should have a superscript a. line 118: are the phi_a additional hyperparameters? How are they set? line 147, 148 "detailed section" -> detailed in Section" line 162: it's not clear to me what you mean by "hierarchically random" line 174: it seems you switched x and z in p(x)=N(z, Sigma). line 177: Gaussian conditioning is not a product. line 181: "the distribution f(b)". This seems a bit odd. Maybe assign f(b) to a random variable? line 189: not sure what you mean by "in principle". - you might want to introduce notation \mathbb{C} and \mathbb{V} line 211: "a mixture of basis functions" -> do you mean it is a linear combination of basis functions? line 245: "physically" -> "to physically" line 273 "collect" -> to collect line 274: trails -> trials line 283: this seems a strong claim, you might want to rephrase (fPILCO doesn't predict how well other methods will do as is implied here ) line 290 - you might want to choose another symbol for noise factor than n (often number of data points). 297 : data - > data points ? 315: Give greater performance than otherwise -> improved performance with respect to the baselines ? References: I would consistently use either full names or initials. Some references like 11 are quite messy (mentioning the year 3 times in this case). The words Gaussian and POMDPs in paper titles should be capitalized correctly.

Reviewer 3



** Summary of Paper The paper proposes an extension to PILCO to improve performance in the presence of noisy observations. The reasoning is that when observations are noisy, the policy should act on the filtered state, not the observations directly. The main contribution of the paper is to not only follow this reasoning during execution of the policy, but also during the prediction/policy evaluation phase of the PILCO algorithm. They argue that this better adapts the policy to acting on the filtered state, allowing it to take more decisive actions. The main challenge that they tackle lies in the fact that they replace observations (i.e. point values) with the filtering distribution. This also means that during prediction, they not only have to predict one distribution over possible future states (due to the probabilistic forward model), but a distribution over possible future noisy observations and the corresponding distribution over future filtering distributions. They evaluate their resulting algorithm on the cartpole swing-up environment. I would recommend this paper for acceptance. It is for the most part well written (see below for details) and proposes a potentially useful extension to a widely used algorithm, even though the set of problems it is applicable to is restricted. While the paper restricts itself to only one environment for experiments, it has two different setups to better show the characteristics of its method. This, together with a clear discussion of strengths and weaknesses of the method, makes it easy to understand whether the proposed algorithm can be useful for one's problem setting. My reasons for not giving a higher rating are 1. The restricted applicability 3. I'm not entirely sure about the fit to NIPS in terms of originality. On the one hand it is 'only' as an extension to an existing algorithm, but on the other hand this extension is non-trivial and requires a considerable amount of thought. ** Quality The paper is technically sound with both a theoretical analysis and convincing experimental results. Strengths and also restrictions of the algorithm are clearly stated and discussed, giving a good understanding of when the algorithm could be useful. ** Clarity The overall quality of the paper is good. For the most part it is well written and easy to follow due to good explanations. However, I think section 4.2 ('Filtered-System Prediction Phase') could be a bit more detailed. While it is possible to understand the explanation, I think it would be easier to do so if it were more explicit in terms of how various random variables influence each other. Some examples/ideas (please correct me if necessary): - In equation (5), I assume \mu^m_{t|t-1} is as known from the previous iteration step? - Also in equation (5), is mu^x_t computed according to the 'standard' PILCO algorithm? The only other mentioning of mu^x is somewhere in equation (1) without explicitly discussing it. Also, I guess instead of /tilde{mu}^x_{t-1} (like in PILCO), one computes it based on mu^{\tilde{m}}? Overall I just think that this section should be at least slightly longer and more detailed as it is describing one of the two core parts of the new method. The following is a very minor point as it's fairly clear what is meant: I am unsure about the notation of the believe b (for example in line 174) as b is sometimes used as if it is a random variable (e.g. line 156) and sometimes as a distribution (e.g. line 175). I think it boils down to (for me), that to me the equation in line 174, namely p(x|z,u) \sim N(m,V), should use an equal sign instead of \sim. A clearer notation here might also resolve that between equations (21) and (22) p(M) and M are used interchangeably? A few small typos I spotted: - Line 7: There's 'with' missing - Line 19: of => for (I think) - Line 127: and => in - Line 147: detailed _in_ section 4.1 ** Originality The proposed algorithm is an extension to a widely used algorithm, building upon several earlier works with a similar goal. It does not propose entirely new concepts (i.e. both filtering and PILCO are widely known) put combines them in a new and non-trivial way. ** Significance The proposed algorithm is useful in a restricted set of problem that are suitable for PILCO but additionally have (close to?) gaussian observation noise. However, given that PILCO is widely used and quite powerful and that the authors of this paper clearly state the strengths and weaknesses of their approach, I believe this to be a valuable contribution.